# Flow-Mediated Vulnerability of Source Waters to Elevated TDS in an Appalachian River Basin

**Eric R. Merriam [1],\*, J. Todd Petty [2] , Melissa O'Neal [3] and Paul F. Ziemkiewicz [3]**

1   Pittsburgh District, U.S. Army Corps of Engineers, Pittsburgh, PA 15222, USA
2   School of Natural Resources, West Virginia University, Morgantown, WV 26506-6125, USA;
    Todd.Petty@mail.wvu.edu
3   West Virginia Water Research Institute, West Virginia University, Morgantown, WV 26506, USA;
    Melissa.O'Neal@mail.wvu.edu (M.O.); paul.ziemkiewicz@mail.wvu.edu (P.F.Z.)
*   Correspondence: eric.r.merriam@usace.army.mil

**Abstract:** Widespread salinization—and, in a broader sense, an increase in all total dissolved solids (TDS)—is threatening freshwater ecosystems and the services they provide (e.g., drinking water provision). We used a mixed modeling approach to characterize long-term (2010–2018) spatio-temporal variability in TDS within the Monongahela River basin and used this information to assess the extent and drivers of vulnerability. The West Fork River was predicted to exceed 500 mg/L a total of 133 days. Occurrence and duration (maximum = 28 days) of—and thus vulnerability to—exceedances within the West Fork River were driven by low flows. Projected decreases in mean daily discharge by ≤10 cfs resulted in an additional 34 days exceeding 500 mg/L. Consistently low TDS within the Tygart Valley and Cheat Rivers reduced vulnerability of the receiving Monongahela River to elevated TDS which was neither observed (maximum = 419 mg/L) nor predicted (341 mg/L) to exceed the secondary drinking water standard of 500 mg/L. Potential changes in future land use and/or severity of low-flow conditions could increase vulnerability of the Monongahela River to elevated TDS. Management should include efforts to increase assimilative capacity by identifying and decreasing sources of TDS. Upstream reservoirs could be managed toward low-flow thresholds; however, further study is needed to ensure all authorized reservoir purposes could be maintained.

**Keywords:** water quality; vulnerability; total dissolved solids; drinking water

## 1. Introduction

Anthropogenic activities are contributing to widespread salinization of streams and rivers [1]. Salinization—and, in a broader sense, an increase in total dissolved solids (TDS)—has important implications for biodiversity [2–4] and ecosystem processes [5]. Elevated TDS also impacts human health and well-being by degrading drinking water quality and/or increasing the cost of water treatment [6], degrading infrastructure, and altering ecosystem goods and services [7].

Elevated TDS concentrations within riverine systems are primarily driven by the spatial pattern and extent of land use change, such as resource extraction, urban development, and agriculture and associated pollutants (acid mine drainage, agricultural and storm water runoff) throughout the watershed [1,8]. Numerous studies have also documented the importance of temporal variability in flow—both natural [9] and anthropogenic [10]—in controlling water quality and its impacts on freshwater ecosystems. Understanding the extent to which temporal variability in flow modulates vulnerability of freshwater ecosystems to elevated TDS across space will be critical to ensuring the resiliency and well-being of individuals and communities that rely on the services they provide [11,12].

This is particularly true given uncertainty regarding the effects of climate change on flow variation, which will likely further impact water quality and alter system vulnerability [13].

Herein, we present results of an analysis of temporal variation in TDS concentrations and its relationship with flow variability in the upper Monongahela River basin in West Virginia, USA. Elevated TDS within the Monongahela River and its major tributaries is primarily the result of historic coal mining and contemporary oil and gas development and has been shown to impact drinking water during low-flow events [14,15]. In 2009, one of the authors (Paul Ziemkiewicz), Director of West Virginia University's Water Research Institute, initiated the Three Rivers Quest (3RQ) program in response to rising TDS levels in the Monongahela River. The result was the identification of treated mine drainage, rich in Ca, Na and $SO_4$ as the controlling factor in the River's TDS load. A discharge management model and agreement among operators of mine discharge treatment facilities was developed and implemented in January 2010. The model allowed the treatment unit operators to modulate discharge load based on flow in the river with allocation of TDS load to ensure that the Monongahela River mainstem would not exceed the secondary drinking water standards for sulfate or TDS (250 and 500 mg/L respectively) [16]. Since then, flows within the Monongahela River and its tributaries inform management decisions, that include timing discharge of treated mine drainage based on assimilative capacity and flow augmentation from upstream reservoirs [17] designed to improve downstream water quality, resulting in additional spatio-temporal variability and complexity in TDS. Consequently, the upper Monongahela River drainage provides a unique and relevant opportunity to characterize how complex spatial and temporal controls over TDS contribute to vulnerability. We use a mixed modeling approach to characterize long-term (2010–2019) spatial and temporal variability within the Monongahela River and its tributaries and use this information to characterize system vulnerability.

## 2. Materials and Methods

### 2.1. Study Area

The study area was defined as the upper Monongahela River drainage upstream of Masontown, Pennsylvania, and drains approximately 11,700 km$^2$ within Pennsylvania and West Virginia. The study area includes the Cheat River, Tygart Valley River, West Fork River, and Upper Monongahela 8 digit hydrologic unit code (HUC) watersheds (Figure 1). The study area is predominately forested (78%). The drainage network is influenced by pre- and post-Surface Mine Control and Reclamation Act (SMCRA) surface mining (3%) and residential and urban development (2%). The stream network also drains 1020 deep mine national pollution discharge elimination system permits, as well as 25,675 conventional and 954 unconventional oil and gas wells [18,19]. Flows within the upper Monongahela River drainage are regulated by two U.S. Army Corps of Engineers reservoirs—Stonewall Jackson Lake on the West Fork River and Tygart Lake on the Tygart River (Figure 1). Both reservoirs were authorized for flood protection and water supply purposes. Water releases from both reservoirs during low-flow periods help maintain commercial navigation and improve water quality for domestic (e.g., drinking water) and ecological (e.g., fish and wildlife habitat) purposes. Flows within the Cheat River are regulated by a private hydroelectric dam (Figure 1).

### 2.2. Data Collection

#### 2.2.1. Water Quality Data

We compiled 969 unique water quality sampling records collected bi-monthly (2010–2014) and monthly (2015–2018) from six sites [three along the Monongahela River (M1, M2, M3), and near the mouths of the Cheat River (CR), Tygart Valley River (TV), and West Fork (WF); Table 1, Figure 1] within the Monongahela River system as part of the Three Rivers Quest water quality monitoring program (https://3riversquest.wvu.edu/).

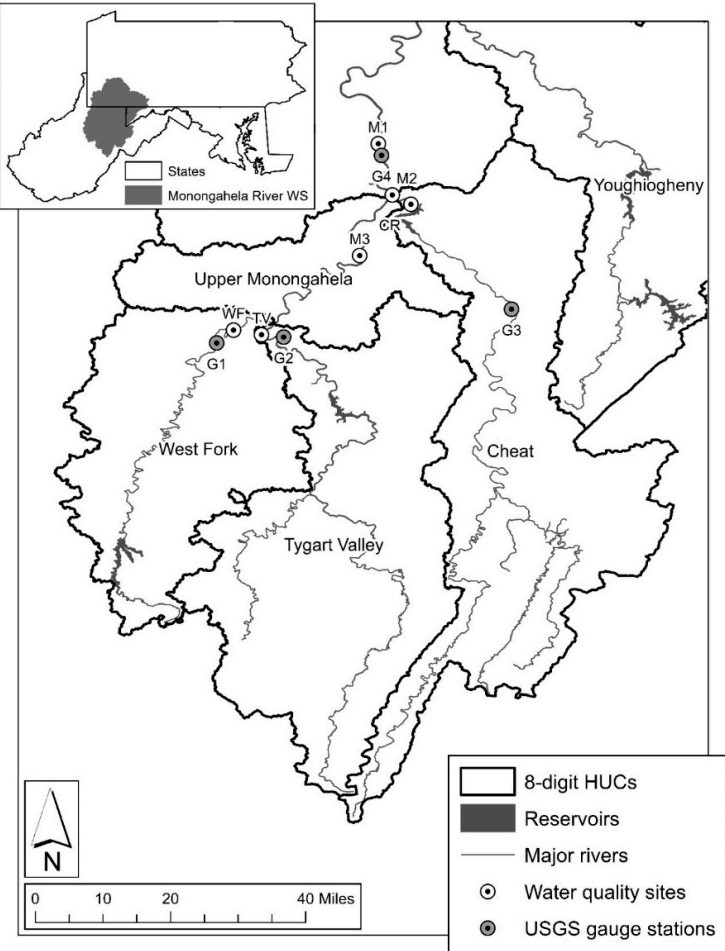

**Figure 1.** Location of water quality sampling sites (n = 6) and United States Geological Survey (USGS) gauging stations (n = 4) used to characterize and predict spatio-temporal variability in total dissolved solids (TDS) within 8 digit hydrologic unit code (HUC) watersheds comprising the upper Monongahela River basin.

**Table 1.** Basin areas, collection dates (month/year), and number of samples (n; days of data) for the six sample sites. Site codes match those presented in Figure 1.

| Site Name | Site Code | BA (km²) | Collection Dates | n |
|---|---|---|---|---|
| Monongahela River 1 | M1 | 11,699 | January 2010–December 2018 | 175 |
| Monongahela River 2 | M2 | 7049 | October 2010–December 2018 | 154 |
| Monongahela River 3 | M3 | 6670 | October 2010–December 2018 | 155 |
| Cheat River | CR | 3645 | October 2010–December 2018 | 155 |
| Tygart Valley River | TV | 3577 | October 2010–December 2018 | 155 |
| West Fork River | WF | 2135 | January 2010–December 2018 | 175 |

Samples were collected and analyzed for dissolved alkalinity (mg/L $CaCO_3$ equivalents; EPA method SM-2320B), dissolved Al, Ca, Fe, Mn, Mg, and Na (mg/L; EPA method 6010B), and dissolved Br, Cl and $SO_4$ (mg/L, EPA method 300.0). Total dissolved solids (TDS) was calculated as the sum of the concentrations of all measured dissolved constituents. In situ measures of temperature, electrical conductivity, and pH were obtained using a YSI 556 multiprobe (Yellow Springs Instruments, Yellow Springs, Ohio).

### 2.2.2. Hydrologic Data

We associated mean daily discharge data from four United States Geological Survey (USGS) gauging stations with the six temporal water quality sites. Four water quality sampling locations [WF (USGS gauge #03061000), TV (03057000), CR (03070260), M1 (03072655)] were associated with individual USGS gauges located on the same stream (Figure 1). We estimated mean daily discharge for the two most upstream sites on the Monongahela River (M2, M3) as the sum of discharges at WF and TV.

### 2.3. Statistical Analyses

#### 2.3.1. Water Quality Characteristics

We calculated summary statistics for observed TDS and its individual constituents to characterize spatial variability among sites and temporal variability within sites.

#### 2.3.2. Water Quality Modeling

We used a hierarchical linear-mixed effect model to predict TDS from mean daily discharge. We log[x] transformed TDS and discharge to approximate normality. We fit a 'beyond optimal model' (BOM) to the training set that allowed slopes and intercepts to vary among sites, years, and years within sites [20]. This random effects structure enabled us to account for site-specific (e.g., upstream land use) and temporal (e.g., annual precipitation) characteristics affecting TDS within the Monongahela River watershed, as well as differences within sites among years (e.g., implementation of regulation or management strategies). We identified the optimal random effects structure by first iteratively dropping random slope terms and comparing the less parameterized model to the BOM [21]. We then iteratively dropped the random intercept terms and compared the less parameterized model back to the optimal random slopes model. We retained random effects with $\Delta$AIC > 2. We assessed model performance by calculating root mean square error (RMSE) and marginal (variance explained by fixed effects) and conditional (variance explained by fixed and random effects) coefficients of determination ($R^2$). We compared observed and expected TDS within the test set to assess predictive accuracy [22]. We used functions in package 'lme4' [23] and 'lmerTest' [24] for model construction and selection. We used package 'MuMIN' to calculate $R^2$ values [25]. We performed all analyses in Program R [26].

#### 2.3.3. Current Conditions and Vulnerability

We applied the optimal model to predict TDS at each sampling location as a function of observed mean daily discharge. We calculated the number of days and number of consecutive days each year with predicted TDS exceeding the EPA secondary drinking water standard of ≥500 mg/L [16]. We calculated the threshold discharge predicted to result in exceedance of the 500 mg/L drinking water standard. We calculated the difference between the threshold and observed mean daily discharge values ($\Delta$ discharge). Positive $\Delta$ discharge values indicate the additional mean daily discharge needed to decrease TDS concentrations below 500 mg/L. Negative $\Delta$ discharge values represent the decrease in mean daily discharge required to result in additional exceedances. We calculated the number of additional days predicted to exceed 500 mg/L when $\Delta$ discharge was ≤100% of the observed value.

## 3. Results

### 3.1. Water Quality Characteristics

Measured water quality was highly variable both within and among study sites (Table 2). Mean observed TDS was highest at WF and exceeded 500 mg/L (maximum = 748 mg/L). TDS was most variable within M1; however, TDS was not observed in excess of 500 mg/L in the Monongahela River (maximum at M1 = 391 mg/L, maximum at M2 = 419 mg/L, and maximum at M3 = 388 mg/L). TDS

was consistently low in in both CR and TV. $SO_4$ was the dominant ion contributing to TDS at all sites. Concentrations of dissolved Al, Fe, and Mn were highest and most variable in WF.

**Table 2.** Means (and standard deviations) of total dissolved solids (TDS) and its contributing constituents across all water samples taken within the Cheat (CR), Tygart Valley (TV), West Fork (WF), and Monongahela (M) Rivers. Refer to Figure 1 and Table 1 for site location and sample information. Means are reported in mg/L. Alkalinity (Alk) is reported in mg/L $CaCO_3$ equivalents.

|  | **M1** | **M2** | **M3** | **CR** | **TV** | **WF** |
|---|---|---|---|---|---|---|
| TDS | 172 (71) | 185 (74) | 176 (69) | 53 (13) | 70 (25) | 363 (127) |
| Alk | 37.2 (15.9) | 44.6 (13.9) | 46.6 (13.6) | 15.4 (6.1) | 25.4 (12.4) | 87.8 (22.1) |
| Br | 0.04 (0.08) | 0.04 (0.09) | 0.02 (0.03) | 0.00 (0.01) | 0.01 (0.02) | 0.05 (0.12) |
| Al | 0.09 (0.11) | 0.10 (0.31) | 0.09 (0.27) | 0.08 (0.13) | 0.05 (0.06) | 0.13 (0.70) |
| Fe | 0.11 (0.15) | 0.16 (0.54) | 0.12 (0.33) | 0.09 (0.16) | 0.08 (0.08) | 0.21 (1.26) |
| Mn | 0.10 (0.07) | 0.09 (0.07) | 0.07 (0.06) | 0.08 (0.06) | 0.05 (0.04) | 0.08 (0.05) |
| Cl | 11.2 (7.1) | 11.2 (4.9) | 9.8 (5.0) | 3.5 (1.4) | 5.5 (2.6) | 15.4 (8.7) |
| Ca | 27.3 (9.1) | 31.2 (9.6) | 30.7 (11.0) | 12.1 (3.7) | 14.6 (3.6) | 65.5 (21.7) |
| Na | 21.1 (13.2) | 20.2 (11.4) | 21.2 (16.2) | 3.3 (2.4) | 7.2 (3.1) | 30.6 (16.8) |
| Mg | 7.0 (2.8) | 8.2 (3.2) | 7.7 (3.1) | 2.6 (1.0) | 3.1 (0.8) | 17.6 (6.5) |
| $SO_4$ | 86 (42) | 91 (45) | 83 (39) | 23 (7) | 27 (18) | 189 (82) |

### 3.2. Water Quality Modeling

The optimal random effects structure for predicting TDS included random intercepts among years and sites, as well as random slopes among sites (Table 3).

**Table 3.** Step-down model selection results for linear-mixed models predicting log[x]-transformed total dissolved solids within the Monongahela River. Asterisks denote random effects retained in the final model (i.e., $\Delta$AIC >2).

| Model Structure | AIC | $\Delta$AIC |
|---|---|---|
| **Beyond optimal model** [a] | 239 | – |
| Dropped random slopes | | |
| Log(Q) \| Year | 239 | 0 |
| * log(Q) \| Site | 257 | 18 |
| Log(Q) \| Site:Year | 249 | −2 |
| Optimal random slope model [b] | 237 | – |
| Dropped random intercepts | | |
| * 1 \| Year | 243 | 6 |
| * 1 \| Site | 400 | 163 |
| 1 \| Site:Year | 235 | −2 |

[a] Beyond optimal model = log(Q) + 1|Site + 1|Year + 1|Site: Year + log(Q)|Site + log(Q)|Year + (Q)|Site: Year. [b] Optimal random slope model = log(Q) + 1|Site + 1|Year + 1|Site: Year + log(Q)|Site.

Mean daily discharge had a significant negative effect on TDS concentration (Table 4). Together, fixed and random effects explained 95% of the total variation in TDS (i.e., marginal $R^2 = 0.95$). Mean daily discharge explained 5% of the overall variation (i.e., conditional $R^2 = 0.05$). The final model had a root mean square error (RMSE) of 0.25 when predicting log[x]-transformed TDS within the test set. We saw a strong relationship ($y = 0.00 + 1.01x$; $R^2 = 0.88$) between observed and predicted TDS within the test set. Predictive accuracy was similar within tributary (RMSE = 0.23) and Monongahela River (RMSE = 0.26) sites.

**Table 4.** Parameter estimates for the best supported model predicting log[x]-transformed total dissolved solids in the Monongahela River.

| Parameter | Estimate | SE | *t*-Value | *p*-Value |
|---|---|---|---|---|
| **Fixed effects** | | | | |
| Intercept | 6.64 | 0.44 | 15.1 | <0.001 |
| Log(Q) | −0.23 | 0.03 | −9.16 | <0.001 |
| Random effects | | | | |
| $\sigma^{1|Year}$ | 0.047 | – | – | – |
| $\sigma^{1|Site}$ | 1.06 | – | – | – |
| $\sigma^{Q|Site}$ | 0.057 | – | – | – |
| $\sigma^{Residual}$ | 0.267 | – | – | – |

*3.3. Current Conditions and Vulnerability*

Predicted TDS was highly variable among sites (Figure 2).

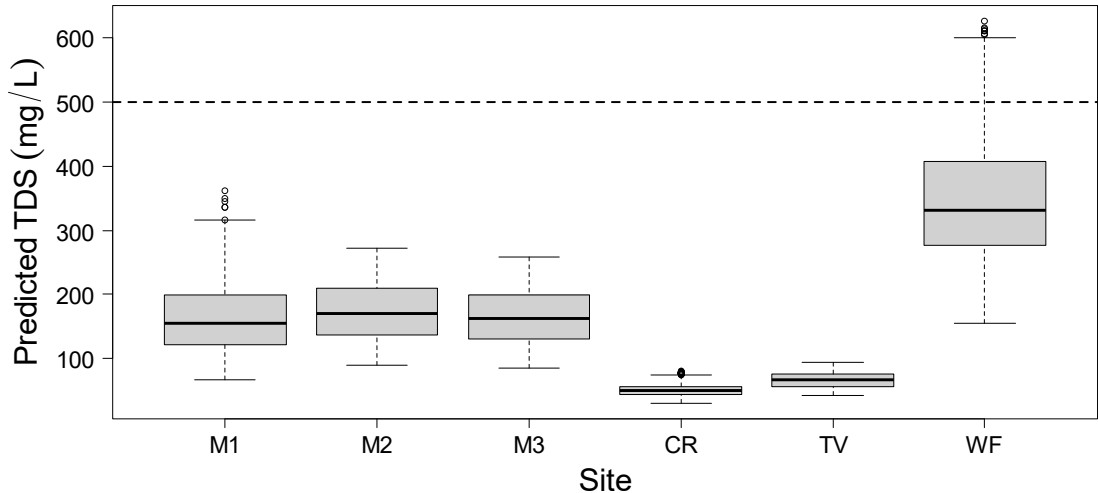

**Figure 2.** Variability in predicted total dissolved solids (TDS) concentrations across all days and years within and among the six sites on the Cheat (CR), Tygart Valley (TV), West Fork (WF), and Monongahela (M) Rivers. Refer to Figure 1 for site locations.

Median TDS was similar across M1 (155 mg/L), M2 (171 mg/L), and M3 (163 mg/L); however, TDS was slightly more variable at M1 (range = 68–362 mg/L) than M2 (90–271 mg/L) and M3 (86–259 mg/L) (Figure 2). Predicted TDS within the Monongahela River never exceeded the EPA secondary drinking water standard of 500 mg/L (Figure 2). Predicted TDS was consistently low within CR (range = 31–80 mg/L, median = 51 mg/L) and TV (range = 43–94 mg/L, median = 67 mg/L) and did not exceed the EPA secondary drinking water standard of 500 mg/L. Predicted TDS was greater (median = 331 mg/L) and more variable (range = 154–626 mg/L) within WF (Figure 2). TDS within WF exceeded 500 mg/L a total of 133 days over 24 events (Figure 3).

Exceedances occurred during six of the nine years, with the number of days exceeding 500 mg/L each year ranging from 2 (2017) to 65 (2010) (Figure 3). The number of consecutive days exceeding 500 mg/L within WF also varied among years and ranged from 1 to 28 (2010) (Figure 3). Inter- and intra-annual variability in exceedances were driven by the frequency and duration of low-flow conditions (Figure 4).

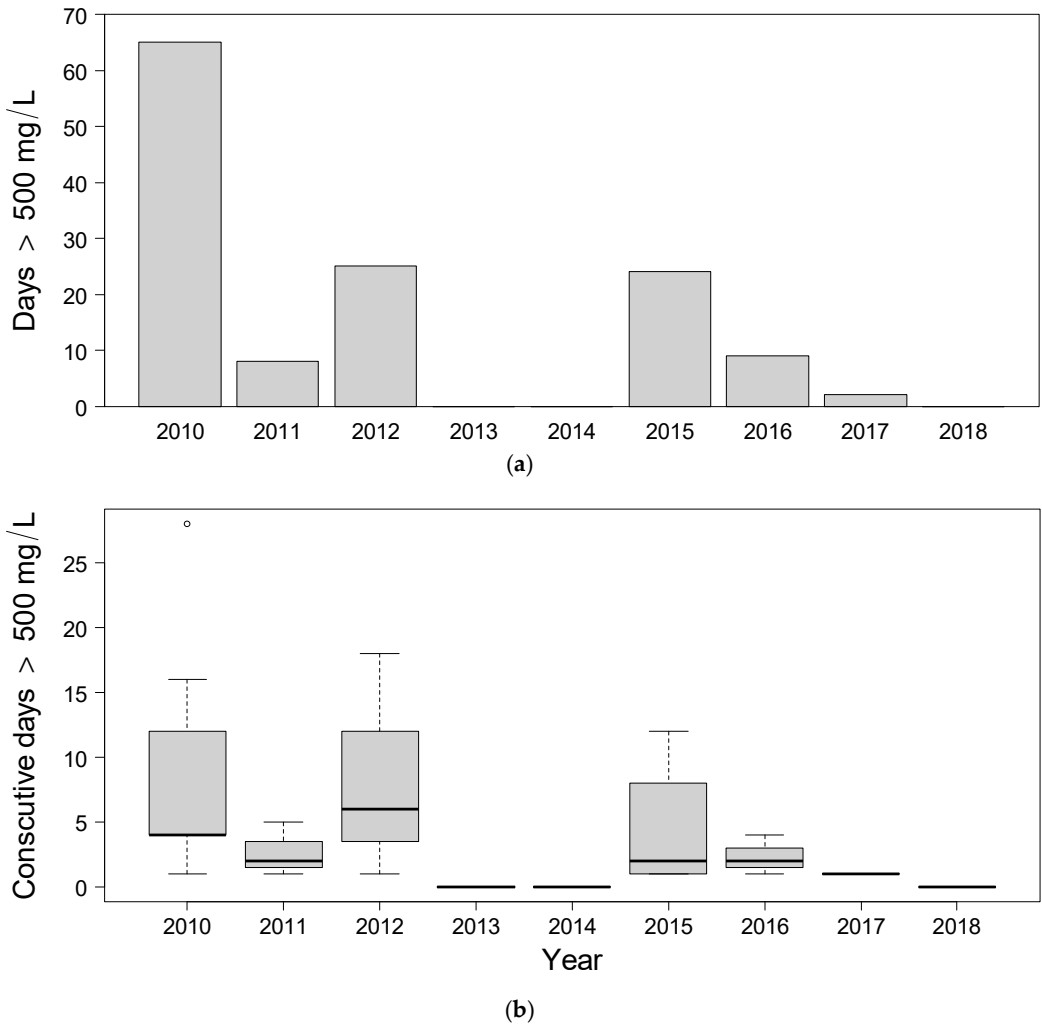

**Figure 3.** (**a**) Histogram showing number of days each year with total dissolved solids (TDS) exceeding 500 mg/L within the West Fork (WF) River; (**b**) Box plots showing the number of consecutive days TDS exceeded 500 mg/L during exceedance events occurring each year.

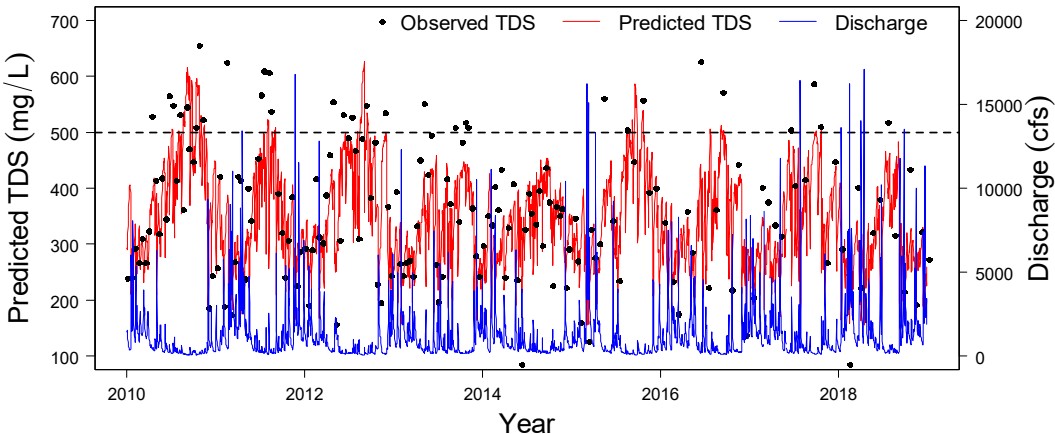

**Figure 4.** Time series of predicted TDS and observed discharge at the West Fork water quality and United States Geological Survey (USGS) gauge station (refer to Figure 1 for locations). Observed TDS is shown for reference.



Threshold discharges predicted to result in exceedance within WF varied among years and ranged from 84 cfs (2013) to 141 cfs (2010). Additional mean daily discharge required to decrease TDS below 500 mg/L (i.e., Δ discharge) during the 24 exceedance events ranged from 1 to 83 cfs (Figure 5).

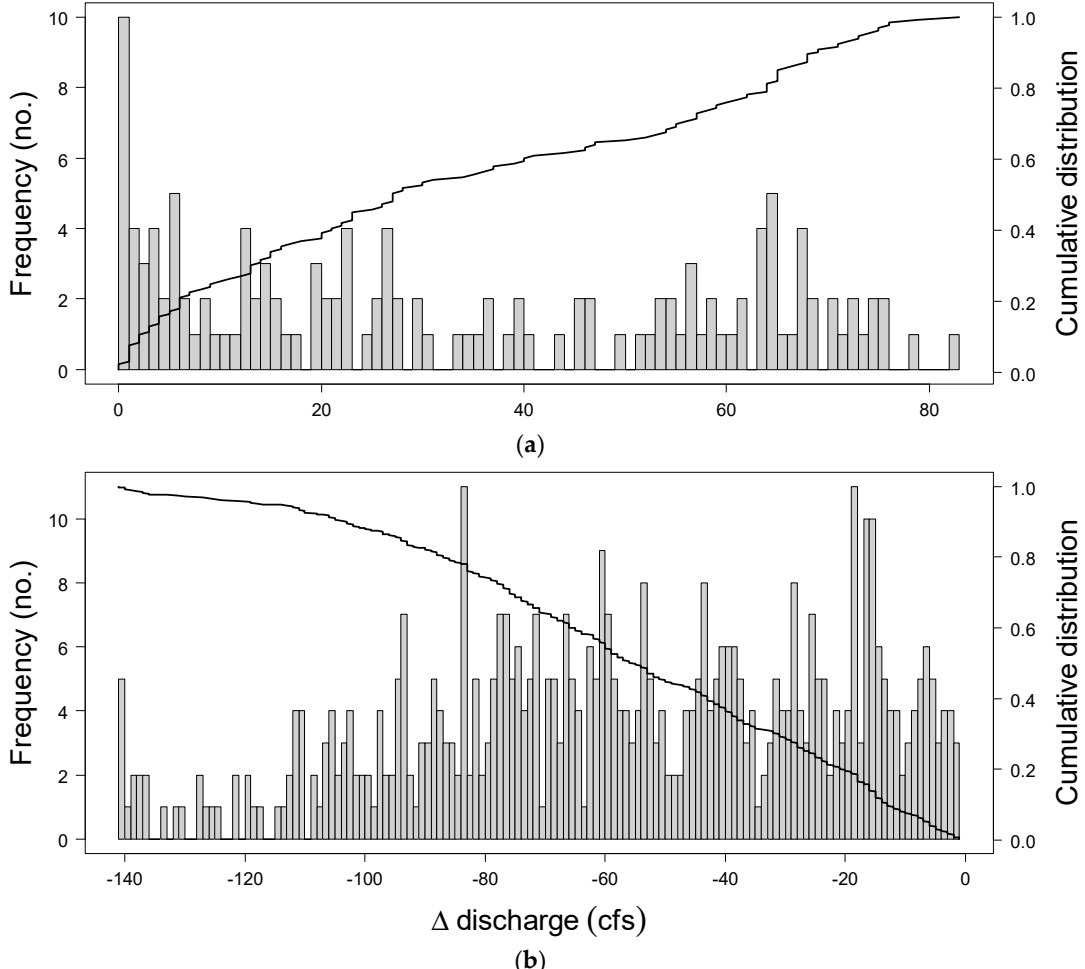

**Figure 5.** (**a**) Frequency and cumulative distribution of increased mean daily discharge (positive Δ discharge) required to decrease predicted TDS below 500 mg/L across the 133 days where the secondary drinking water standard was exceeded; (**b**) Frequency and cumulative distribution of decreased mean daily discharge (negative Δ discharge) required to result in exceedance of the 500 mg/L across the 499 days with Δ discharge < 100% of the observed value.

Mean daily discharge within WF fell short of the exceedance threshold by ≤1 cfs 10 times (8% of total exceedances). Twenty-four percent (26%; 34 days) of exceedances were characterized by Δ discharge ≤10 cfs (Figure 5). A total of 499 additional days were predicted to exceed 500 mg/L with simulated decreases in mean daily discharge (Δ discharge) ≤100% of the observed value (Figure 5). Of those, 39 days (8%) were predicted to exceed 500 mg/L with a decrease in observed discharge of ≤10 cfs (i.e., Δ discharge ≥−10) (Figure 5).

## 4. Discussion

We characterized a high degree of spatial and temporal variability in TDS that provided critical insight into vulnerability of the upper Monongahela River basin to elevated TDS. TDS within the West Fork River (WF) was predicted to exceed the secondary drinking water standard of 500 mg/L a total of 133 days from 2010 to 2018, with the frequency and duration of exceedances being closely tied to low-flow conditions. Consistently low TDS within the Tygart Valley River (TV) and Cheat River (CR)

reduced vulnerability of the Monongahela River (M1, M2, M3) to elevated TDS, which was neither observed (maximum = 419 mg/L) nor predicted (341 mg/L) to exceed 500 mg/L.

Dominance of $SO_4$ in the ionic signature and variability in elevated Al, Fe, and Mn suggest and support previous studies identifying mining as the main contributor to elevated TDS in the Monongahela River basin [27,28]. Elevated concentrations of and variability in other major ions (Ca, Mg, Na, Cl) could be attributed to other land use activities such as oil and gas development [14,15] and urban development [27]. The extent to which these landscape features contribute to elevated TDS ultimately controls vulnerability of the system to variability in flow. TDS concentrations within CR and TV remained below 500 mg/L regardless of flow, suggesting limited vulnerability to variability in flow. In contrast, TDS concentrations within WF were elevated to the point that temporal variability in TDS associated with changes in flow were enough to result in exceedance of the 500 mg/L drinking water criterion. These results corroborate previous work documenting the importance of flow variability—both natural (e.g., precipitation patterns [9]) and anthropogenic (i.e., discharge management and reservoir releases [10])—in modulating vulnerability of this and other freshwater systems to degraded water quality [12,29].

TDS was predicted to exceed the 500 mg/L drinking water criterion a total of 133 times within the West Fork River. The number of consecutive days with predicted TDS exceeding 500 mg/L ranged from 1 to 28. TDS concentrations exceeding 500 mg/L can lead to pipe corrosion, increased scaling and sedimentation, and taste and odor problems [15], potentially increasing the cost of water treatment and maintenance of degraded infrastructure. Thus, TDS concentrations observed and predicted in the current study have important implications for the individuals and communities obtaining their drinking water from the West Fork River (e.g., 60,000 individuals serviced by a single water supplier). Despite these concerns, TDS is not specifically targeted for removal during conventional drinking water treatment, nor is the 500 mg/L secondary drinking water standard mandatory or enforceable [15,16]. Consequently, it is incumbent upon water resource agencies to manage TDS in an effort to help ensure continuity of vital ecosystem services within the West Fork and Monongahela Rivers both now and into the future.

Programs like 3RQ will be most successful if done within the context of flow variability and assimilative capacity. The threshold discharge (i.e., discharge resulting in exceedance of the 500 mg/L) identified for the West Fork River in the current study was higher in 2010 (141 cfs) than subsequent years (84–140 cfs), suggesting increased assimilative capacity in the West Fork and Monongahela Rivers following implementation of the discharge management model. This assertion is further supported by a predicted decrease in the frequency and duration of drinking water standard (i.e., 500 mg/L) exceedances within the West Fork River following 2010. Defining a pollutant's chemical footprint—the volume of water available to dilute additional pollution loading or required to dilute existing loads to allowable levels (e.g., TDS of 500 mg/L)—represents another tool for managing water quality within the context of assimilative capacity [30,31]. This information can then be used to strategically target management efforts (i.e., reducing current pollution loads and/or protecting areas for maintenance of downstream assimilative capacity) that maximize system resiliency and decrease vulnerability.

Our results could also be used to inform updated management of the Stonewall Jackson and Tygart River Reservoirs. Augmenting flows through additional reservoir releases during critical low-flow periods could decrease current and future vulnerability of the West Fork and Monongahela Rivers to elevated TDS. The minimum flow thresholds required to maintain TDS below 500 mg/L identified in this study could inform such an effort and be incorporated into an adaptive management plan [32]. Both reservoirs currently augment low flows to maintain downstream water quality and are vital to the maintenance of drinking water standards in both rivers. However, any effort to alter reservoir releases must not affect the reservoirs' capacity to maintain all authorized purposes (i.e., flood protection, water supply for maintenance of water quality and navigation, recreation, and fish and wildlife enhancement) over both the short- and long-term. Additional research is needed to characterize how altering reservoir releases would affect all ecological (e.g., reservoir and downstream

water quality and habitat) and socioeconomic (e.g., recreation, water supply) systems under both realized and potential future low-flow conditions.

Our study represents one of only a few to assess how spatio-temporal variability in TDS concentrations contribute to vulnerability of freshwater ecosystems and how they should be managed (e.g., drinking water) (but see [11,12]). Our results corroborate previous studies documenting increased vulnerability of surface waters to elevated ionic concentrations during periods of low flow [12]. Salts are generally transported to streams via subsurface flow through natural or altered (i.e., mined) landscapes. Ionic concentrations are generally greatest during dry periods when stream flow is dominated by subsurface flow and becomes diluted with increasing surface runoff [33]. In contrast, streams and rivers are often most vulnerable to pollutants transported to streams via surface runoff (e.g., nutrients) during high flow events [12]. Spatial and temporal complexities make managing toward individual pollutant criteria difficult [11]. Spatial and temporal complexities among multiple pollutants make maintaining functional ecosystems and ecosystems services difficult. Future research should focus on characterizing spatio-temporal variability of key pollutants and how that variability controls vulnerability of critical source waters.

The methodology described herein can be used to assess current and future vulnerability of, and create management plans for, systems impacted by any number of pollutants. The mixed modeling framework used in this study enabled us to quantify and predict the effects of discharge (i.e., fixed effects), while accounting for site- (e.g., upstream land use) and year-specific (e.g., temporal changes in water quality management) factors that affect spatio-temporal patterns in TDS and response to flow variability. This approach enabled us to predict TDS concentrations with a high degree of certainty and accuracy ($R^2$ = 0.95; test set RMSE = 0.25). Our study also highlights the value and utility of long-term monitoring data for providing insight into continuous water quality conditions in systems where such data are unavailable.

Our results suggest the upper Monongahela River basin may be vulnerable to even minor changes in TDS and/or discharge. Decreases in mean daily discharge by ≤10 cfs resulted in an additional 34 days exceeding 500 mg/L within the West Fork River. Although the Monongahela River was never observed or predicted to exceed 500 mg/L, potential changes in future land use impacts (e.g., continued expansion of unconventional oil and gas, additional mine drainage contributions or reduced treatment of current mine discharge) and/or discharge could elevate TDS levels to the point that this system becomes vulnerable to variability in flow. Elevated temperatures (i.e., increased evapotranspiration [33]) and increased variability in precipitation under climate change have the potential to exacerbate drought and low-flow conditions, further reducing assimilative capacity within the upper Monongahela River basin during critical low-flow periods [34–36]. In that event, the discharge management model would require adjustment to account for changes in either TDS load or assimilative capacity. Climate change is also expected to interact with and amplify hydrologic modifications in the built environment (e.g., increased stormwater runoff, reservoir systems), as well as other anthropogenic activities (e.g., mine drainage and increased water extraction) to further impact water quality and associated services [13,37–39]. Given the role flow variability plays in modulating vulnerability of receiving waters to TDS, uncertainty regarding the effects of climate change on flow within this region makes this an area of concern. This is particularly true given the role Appalachian headwater streams will play in contributing to and securing regional water supply throughout the 21st century [40].

## 5. Conclusions

Results of the present study demonstrate the myriad of spatial (e.g., land use) and temporal (e.g., variability in flow) factors that control vulnerability of source waters to elevated TDS. Management decisions that do not incorporate these complexities risk ineffective or inappropriate actions. Management of TDS within this and other systems should first seek to identify and prioritize areas for reducing current sources of TDS throughout the watershed and for protection of minimally impacted streams to maintain current and future assimilative capacity. It could also be possible to

leverage existing water management and reservoir systems to maintain assimilative capacity during critical low-flow periods; however, additional study would be needed to verify that these reservoirs have the capacity to maintain all authorized purposes (e.g., navigation, fish and wildlife habitat enhancement) while providing additional low-flow augmentation for maintenance of water quality. Given widespread salinization of streams and rivers [1], an important avenue of continued research will be to characterize the spatio-temporal vulnerability of this and other critical source waters to elevated TDS. It will be particularly important to characterize vulnerability to TDS within the context of other pollutants and under a range of future land use and climate change (i.e., flow variability) scenarios. Such efforts will be critical toward effectively ensuring sustainability of aquatic ecosystems and the vital services they provide (e.g., drinking water provision).

**Author Contributions:** Conceptualization, E.R.M. and J.T.P.; Formal Analysis, E.R.M.; Investigation, M.O.; Resources, P.F.Z.; Data Curation, M.O.; Writing—Original Draft Preparation, E.R.M.; Writing—Review and Editing, J.T.P., P.F.Z., and M.O.; Project Administration, E.R.M., J.T.P, and M.O.; Funding Acquisition, J.T.P., P.F.Z., M.O., and E.R.M. All authors have read and agreed to the published version of the manuscript.

**Funding:** This research was funded by the National Science Foundation, award number OIA-1458952. This work was partially supported by the U.S. Army Corps of Engineers Planning Assistance to States Program. Any opinions, findings, and conclusions or recommendations expressed in this material are those of the authors and do not necessarily reflect the views of the National Science Foundation or U.S. Army Corps of Engineers. Funding in support of the 3RQ program was provided by the Colcom Foundation, Pittsburgh PA and through the U.S. Geological Survey 104b program.

**Acknowledgments:** Field sampling by West Virginia University staff: Jason Fillhart, Benjamin Mack, Benjamin Pursglove; undergraduate students: Kaylynn Kotlar, Reva Dickson, Joshua Ash, Anthony Diamario, Jason Eulberg, Tyler Richards, Jude Platz, Cullen Platz, and Alex Pall; graduate students: Eric Baker, Zac Zacavish, Chance Chapman, Madison Cogar, Joseph Kingsburg, and Levi Cyphers. The authors would also like to thank Rosemary Reilly (Pittsburgh District, U.S. Army Corps of Engineers) and two anonymous reviewers for providing comments that greatly improved the quality of the manuscript.

**Conflicts of Interest:** The authors declare no conflict of interest.

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
