# Peer review of "Flow-Mediated Vulnerability of Source Waters to Elevated TDS in an Appalachian River Basin"

_water, doi:10.3390/w12020384_

Round 1

Reviewer 1 Report

Fig 2 presents spatial variability, no presents temporal variability. What are reasons for the very high TDS in West Fork. Across the 139 days and 499 days. On what basis were these periods determined. In the discussion make a comparison to the results of other authors.

Author Response

RESPONSE TO COMMENTS

REVIEWER #1

Fig 2 presents spatial variability, no presents temporal variability. The reviewer is correct in that this figure uses box plots to primarily show variability among sites. However, the box plots themselves reflect variability in predictions throughout the entire study period. Therefore, temporal variability in predicted TDS within each site is reflected in the structure of each boxplot. For example, predicted TDS exhibits much less temporal variability (i.e., smaller box plot) as compared to WF (i.e., much wider box plot with a greater distribution of predicted values). We do not feel that any change is required to the manuscript; however, we will default to the managing editor if they believe a change is warranted.

 What are reasons for the very high TDS in West Fork. We appreciate this comment and attempt to more clearly show and discuss what the drivers of TDS are throughout the study area. Specifically, we now summarize all water quality constituents contributing to elevated TDS (new Table 2). We then discuss these results in the second paragraph of the discussion (paragraph from lines 218 – 230 of the revised manuscript). These results suggest that mining is the major contributor to elevated TDS (i.e., dominance of SO4); however, variability in other major ions (Ca, Mg, Cl, Na) suggest other land use activities are contributing to elevated TDS. We note that these results corroborate other studies characterizing TDS sources within the watershed.

 Across the 139 days and 499 days. On what basis were these periods determined. 133 days reflects the number of days where predicted TDS exceeded the 500 mg/L secondary drinking water standard. The methodology for this is described in lines 129–130 of the revised manuscript. These results are presented in lines 179–180 of the revised manuscript.

499 days reflects the number of days predicted to exceed the 500 mg/L criterion when simulated flows were <100% of the observed value. The methodology for this is described in lines 135–136 of the revised manuscript. These results are presented in lines 206–209 of the revised manuscript.

We feel that the methodology and associated results are presented clearly both within the text and within figure captions. Therefore, no additional changes were made. We will defer to the managing editor on whether additional changes are needed.

 In the discussion make a comparison to the results of other authors. We appreciate this comment and bolstered several sections of the discussion to more thoroughly compare to existing literature. These changes can be found in lines 218–222 with respect to sources of TDS and lines 269–281 with respect to the importance of spatio-temporal variability in driving vulnerability to degraded water quality.

 REVIEWER #2:

The matter of the manuscript Flow-mediated vulnerability of source waters to elevated TDS in an Appalachian River basin is vital and fits into the scope of the Water journal. The results presented are relevant and thought-provoking, which deserves publication. The experiment procedure and analysis are generally correct. The research scale is really impressing – 969 unique water quality sampling records from 6 sites provide the solid basis for further data interpretation. We appreciate this comment and are glad to see the reviewer finds the study to be of interest to the readers of ‘Water’.

 As the manuscript provides the data on the measurement of TDS in water, my key question is rather simple: what are the implementation prospects of your findings? You suggest certain ways, including strategically target management efforts. If the results are of core value for basic research of natural water properties, it should be emphasized in the conclusion too. We appreciate this comment and expanded the discussion of how our results fit within the context of existing research assessing how spatio-temporal variability drives vulnerability to degraded water quality. We summarize with key areas of future research. These changes can be found in lines 269–281 of the revised manuscript. We also added additional text summarizing future research needs in lines 318–326 of the conclusion.

 Is it possible to provide the data on treated mine drainage composition? This can be interesting, especially if mining is one of the anthropogenic landscape characteristics contribute to spatial variability in elevated TDS throughout the upper Monongahela River basin (l. 200–202). We agree that providing additional data on the components of TDS within our sites would help the reader conceptualize causes and potential research/management needs. Therefore, we included summary statistics for observed TDS and its constituents in a new table (Table 2 of the revised manuscript). We then discuss these results within the context of existing literature in lines 218 – 222 of the discussion.

Reviewer 2 Report

The matter of the manuscript Flow-mediated vulnerability of source waters to elevated TDS in an Appalachian River basin is vital and fits into the scope of the Water journal. The results presented are relevant and thought-provoking, which deserves publication. The experiment procedure and analysis are generally correct. The research scale is really impressing – 969 unique water quality sampling records from 6 sites provide the solid basis for further data interpretation

As the manuscript provides the data on the measurement of TDS in water, my key question is rather simple: what are the implementation prospects of your findings? You suggest certain ways, including strategically target management efforts. If the results are of core value for basic research of natural water properties, it should be emphasised in the conclusion too.

Is it possible to provide the data on treated mine drainage composition? This can be interesting, especially if mining is one of the anthropogenic landscape characteristics contribute to spatial variability in elevated TDS throughout the upper Monongahela River basin (l. 200–202).

I sincerely hope you will find my suggestions helpful.

Kind regards,

Reviewer

Author Response

(The authors gave the same response as above.)
